# Factors influencing dignity impairment in elderly patients with incontinence-associated dermatitis: A lasso and logistic regression approach

**Jinlei Du**[1]*, **Xiaoling Wu**[2], **Ling Lei**[2], **Hongxiang Zhao**[2], **Qiyu Zhang**[2], **Yuanxia Wang**[2], **Yao Chen**[2], **Chencong Nie**[2], **Jiquan Zhang**[3]

**1** Sichuan Vocational College of Health and Rehabilitation, Zigong, Sichuan Province, China, **2** Zigong Fourth People's Hospital, Zigong, Sichuan Province, China, **3** Deyang People's Hospital, Deyang, Sichuan Province, China

\* dujinlei1128@163.com

## Abstract

### Objective

To understand the current status of dignity impairment in elderly patients with incontinence-associated dermatitis and to systematically analyze the factors influencing it.

### Methods

This study employed a cross-sectional survey design. From July 15th to 31st, 2024, a convenience sampling method was used to select elderly patients with incontinence-associated dermatitis (IAD) from 42 tertiary comprehensive medical institutions across 11 provinces and autonomous regions, including Sichuan, Zhejiang, and Guangdong. A general information questionnaire and a dignity assessment scale were used to gather data on participants' demographic characteristics, clinical conditions, and dignity impairment symptoms.

### Results

A total of 372 patients were effectively surveyed, of whom 131 exhibited symptoms of dignity impairment, resulting in an incidence rate of 35%. Multifactorial analysis revealed that Employment Status [$OR$=4.183, 95% CI (2.233-7.837)], Primary Caregiver [$OR$=1.451, 95% CI (1.005-2.095)], Respiratory System Disease [$OR$=5.053, 95% CI (2.079-12.279)], and Nervous System Disease [$OR$=2.452, 95% CI (1.206-4.985)] were risk factors for dignity impairment symptoms in elderly patients with incontinence-associated dermatitis. Conversely, Gender [$OR$=0.488, 95% CI (0.295-0.807)] and Self-Reported Family Harmony [$OR$=0.703, 95% CI (0.550-0.898)] were identified as protective factors against dignity impairment symptoms in this patient population.

### Conclusion

Although the incidence of dignity impairment symptoms in elderly patients with incontinence-associated dermatitis is relatively low, the psychological trauma it causes cannot be

**Data availability statement:** The data supporting the findings of this study are available in the Zenodo repository at https://doi.org/10.5281/zenodo.14903929.

**Funding:** This study was supported by the Sichuan Vocational College of Health and Rehabilitation (CWKY-2024Y-08).

**Competing interests:** The authors have declared that no competing interests exist.

overlooked. Healthcare professionals must actively establish and enhance prevention and management systems for this condition.

## 1 Introduction

Incontinence-associated dermatitis (IAD) is a form of contact irritant dermatitis caused by exposure to urine or feces, commonly occurring in patients with urinary or fecal incontinence [1]. This condition can affect skin areas beyond the perineum [2]. Research reports indicate that more than 200 million people worldwide suffer from severe urinary and fecal incontinence, with the incidence of incontinence-associated dermatitis ranging from 15.8% to 50.0%, particularly prevalent among elderly patients [3]. Studies indicate that due to skin aging, thinning of the epidermis, and decreased elasticity, the incidence of IAD in elderly patients can be as high as 47.7% [4]. As global aging intensifies, IAD in elderly patients has become a prominent health issue, significantly affecting their quality of life while increasing medical costs and caregiving burdens, thus causing considerable physical and psychological distress [5].

IAD is a common yet under-researched condition in elderly patients, particularly in relation to the psychological impact it has on their dignity. While current research focuses largely on the prevention and treatment of IAD symptoms, limited attention has been given to the associated psychological aspects, particularly the loss of dignity, in older adults [6].Dignity, a fundamental right encompassing respect, recognition, and fair treatment, holds significant value in determining an individual's well-being, especially for elderly individuals who may experience compromised health and functionality [7].

When dignity is compromised, it can lead to severe consequences on mental health, disease prognosis, and overall quality of life [8]. Elderly individuals with IAD may experience significant distress due to enforced postures, exposure of personal privacy, and restraint management, which exacerbate feelings of personal space invasion and rights infringement [9]. These psychological factors are often exacerbated by their conservative beliefs and cultural values, making it essential to address dignity loss as part of the comprehensive management of IAD [10].

By incorporating these psychological factors into the clinical management of IAD, we aim to improve both the physical and emotional well-being of elderly patients. This study, utilizing Lasso regression for univariate screening followed by binary logistic regression, identifies key factors contributing to dignity impairment in elderly patients with IAD, providing valuable insights for improving both their physical care and psychological well-being.

## 2 Materials and methods

### 2.1 Study population

From July 15 to July 31, 2024, this study employed a convenience sampling method to recruit elderly patients with incontinence-associated dermatitis (IAD) from 42 tertiary comprehensive hospitals located in 11 provinces and municipalities, including Sichuan, Zhejiang, Guangdong, Guiyang, Hunan, Shanghai, Hubei, Fujian,Shanxi, Anhui and Inner Mongolia. Prior to the main study, a preliminary survey involving 40 patients meeting the inclusion and exclusion criteria was conducted in our city. This survey indicated a prevalence rate of dignity impairment among elderly IAD patients of approximately 30%.

The required sample size for this study was calculated using the formula $N = \frac{Z^2 . P . (1-p)}{E^2}$ , where N represents the needed sample size, Z is the z-score corresponding to the desired confidence level (Z = 1.96 for a 95% confidence interval), P is the estimated prevalence rate, and E is the allowable margin of error, set at 0.05. Based on these parameters, the calculated sample

size was 323. Considering a 10% attrition rate, the final required sample size was adjusted to 355. Ultimately, data were collected from 372 valid participants.The slight increase in the final sample size was primarily aimed at ensuring the stability and reliability of the study results. By including a slightly larger sample, we minimized the potential impact of random variations and missing data, which enhances the robustness of our findings. This approach also strengthens the statistical power of the analysis, especially in addressing potential variations across different subgroups, and ensures that our conclusions are both reliable and generalizable.

Inclusion criteria: Age ≥60 years; diagnosis of incontinence-associated dermatitis according to international diagnostic and grading guidelines [11]; voluntary participation with signed informed consent; Fully conscious and able to communicate without impairment.

Exclusion criteria: History of depression within the past three months; diagnosis of psychiatric disorders; presence of other severe, incurable chronic diseases such as advanced cancer, end-stage liver or kidney disease, or any other terminal illness;presence of severe physical disabilities, such as complete dependence on others for daily activities;presence of other dermatological conditions, including infectious skin diseases (e.g., cellulitis) or autoimmune skin disorders (e.g., psoriasis or systemic lupus erythematosus);participation in concurrent psychological intervention studies.

## 2.2 Ethical consideration

This project conforms to the National Statement on Ethical Conduct in Human Research (2016) and has been approved by the Ethics Committee of Zigong Fourth People's Hospital (approval number: EC-2023-073). The study was conducted in accordance with the Declaration of Helsinki. In this study, all participants signed informed consent forms before the research was conducted. All interviews and surveys were conducted in private settings to ensure participant comfort and confidentiality. Participants were informed that their participation was entirely voluntary and that they could withdraw from the study at any time without any consequences. All collected information was processed anonymously to protect participant privacy.

## 2.3 Survey tools

**2.3.1 General information questionnaire.** The research team designed the questionnaire based on a review of literature, expert consultations, and qualitative interviews with patients. The questionnaire consisted of general demographic information and disease-related data.

General demographic information included: sex, age, educational level, marital status, personality traits, occupation, payment method for medical expenses, primary caregiver, and self-assessed family harmony. Marital status categories for "Non-married" include divorced, widowed, or never married. Occupation categories included "Staff of public institutions," referring to civil servants, teachers, etc. Additionally, the "other" category for methods of medical expense payment includes out-of-pocket expenses and commercial insurance. Disease-related data included: primary disease, duration of incontinence-associated dermatitis, and the grading of incontinence-associated dermatitis, which was classified according to the 2015 International Expert Consensus on Incontinence-Associated Dermatitis as follows: Level 0 (skin intact, no erythema), Level 1 (skin intact, erythema present, with or without edema), and Level 2 (erythema with skin breakdown, edema, vesicles, erosions, or infection).

**2.3.2 Dignity inventory.** The Dignity Inventory was initially developed by Canadian scholar Chochinov [12] in 2002 and formally published in 2008. This inventory includes five dimensions: symptom distress, psychological status, dependency, social support, and spiritual well-being, comprising a total of 25 items. Patients self-assess their symptoms, ranging from no distress to severe distress, on a scale from 1 to 5 for each item. Each item is scored from 1

to 5, and the total score of the inventory ranges from 25 to 125. When the total score of the inventory exceeds 50, it indicates the presence of significant dignity impairment symptoms in patients. The reliability and validity of this scale have been reported as follows: Cronbach's α coefficient is 0.924, and content validity is 0.885 [13].

## 2.4  Data collection methods

Before the study commenced, project team members provided detailed introductions to the research background and survey methods to doctors or nurses within each medical institution. During the study period, the project leader distributed the survey questionnaires to various medical institutions via mail or delivery. Patients independently completed the questionnaires and submitted them anonymously. Each patient could only fill out the questionnaire once, and all contents had to be completed before submission. After all data collection was completed, responsible personnel from each region collected and mailed the questionnaires uniformly.

## 2.5  Quality control

To ensure the accuracy of data from the multi-center survey, the research team established a dedicated research group prior to the study initiation. Additionally, a survey coordinator was appointed at each hospital and underwent standardized training. These coordinators were responsible for quality control and data collection within their respective hospitals.

All survey questionnaires were jointly reviewed and screened by two researchers. Questionnaires with entries showing maximum or minimum values or evidence of repeated modifications, which posed a risk of bias, were excluded. Furthermore, the analysis and processing of survey data were reviewed and guided by experts with a statistical background from our institution to ensure the accurate interpretation of statistical results and their clinical significance

## 2.6  Statistical methods

Data entry with double-checking of questionnaires was conducted using Epidata 3.0 data processing software, and original data were statistically analyzed using SPSS 24.0 and R software. For count data, frequency descriptions were utilized. Continuous data were described using mean ± standard deviation for normally distributed variables and median with inter-quartile range for non-normally distributed variables.

To enhance the robustness of our variable selection process, we employed LASSO (Least Absolute Shrinkage and Selection Operator) regression analysis. LASSO was used to refine our variable selection by applying an L1 penalty to the regression coefficients, performing both variable selection and regularization. This method identifies the most significant predictors of dignity impairment symptoms by shrinking the coefficients of less important variables to zero. Variables with non-zero coefficients in the LASSO model were then included in a binary logistic regression analysis to evaluate their impact on dignity impairment symptoms among elderly patients with incontinence-associated dermatitis. In the univariate analysis, variables were selected based on their clinical relevance and statistical significance ($P < 0.05$). Prior to performing the logistic regression analysis, we conducted collinearity diagnostics using variance inflation factors (VIF) to ensure that selected variables were not highly correlated. Any variable with a VIF greater than 10 was excluded to avoid multicollinearity. To assess the model fit, we used the Hosmer-Lemeshow goodness-of-fit test, which divides the data into deciles based on predicted probabilities and compares observed and expected frequencies within each decile. A non-significant p-value ($p > 0.05$)

indicates that the model fits the data well. Throughout all analyses, statistical significance was defined as P < 0.05.

## 3  Results

### 3.1  General information of survey participants and LASSO-based variable selection for dignity impairment symptoms

In this study, a total of 381 elderly patients with incontinence-associated dermatitis who met the inclusion and exclusion criteria were invited to participate in the questionnaire survey. Among them, 5 questionnaires had responses that were either maximum or minimum values, and 4 questionnaires showed signs of repeated erasures, and were thus excluded. Ultimately, we obtained 372 valid responses, resulting in an effective response rate of 97.63%. Among the 372 patients, the dignity assessment score was $44.45 \pm 19.63$. Among them, 131 patients experienced dignity impairment symptoms, with a total score on the inventory of $74.26 \pm 12.79$. Specifically, the scores for different dimensions were as follows: psychological status dimension: $14.55 \pm 5.77$, symptom distress dimension: $19.09 \pm 6.17$, spiritual well-being dimension: $11.11 \pm 6.60$, dependency dimension: $22.16 \pm 2.87$, and social support dimension: $7.08 \pm 2.14$.

Among the 241 patients who did not experience dignity loss symptoms, the total score on the inventory was $33.37 \pm 8.15$. Specifically, the scores for different dimensions were as follows: psychological status dimension: $8.01 \pm 4.29$, symptom distress dimension: $5.64 \pm 2.65$, spiritual well-being dimension: $6.44 \pm 1.72$, dependency dimension: $7.75 \pm 3.29$, and social support dimension: $5.22 \pm 2.23$.

The results of the LASSO (Least Absolute Shrinkage and Selection Operator) analysis showed that several variables were significantly associated with the assessment of dignity impairment symptoms in elderly patients with incontinence-associated dermatitis. The variables selected by LASSO for further analysis include gender, age, place of residence, primary occupation, employment status, primary caregiver, living situation, self-reported family harmony, primary disease and grading of IAD.Please refer to S1 Table. Variables Selected by LASSO Analysis for Assessing Dignity Impairment Symptoms in Elderly Patients with Incontinence-Associated Dermatitis, and S1 Fig. Lasso Coefficient Path for Variables in Assessing Dignity Impairment Symptoms in Elderly Patients with Incontinence-Associated Dermatitis for details.

### 3.2  Multivariate analysis results: impact of key predictors on dignity impairment symptoms based on LASSO-selected variables

Using the assessment results of dignity impairment symptoms in elderly patients with incontinence-associated dermatitis as the dependent variable, the independent variables selected through LASSO analysis were included in the regression model for multifactor logistic regression analysis. Before conducting the multifactorial analysis, collinearity diagnostics were performed on the variables with statistical differences in the univariate analysis. It was found that the tolerance of each model was > 0.1, and the variance inflation factor was less than 10, indicating that there was no multicollinearity between the variables [14].

Multifactor logistic regression analysis showed that Employment Status [OR = 4.183,95%CI = (2.233-7.837)],Primary Caregiver [OR = 1.451, 95%CI = (1.005-2.095)],Respiratory system disease [OR = 5.053,95%CI = (2.079-12.279)] and Nervous system disease [OR = 2.452, 95%CI = (1.206-4.985)] were risk factors for dignity impairment symptoms in elderly patients with Incontinence-Associated Dermatitis.Gender [OR = 0.488,95%CI = (0.295-0.807)] and Self-Reported Family Harmony [OR = 0.703, 95%CI = (0.550-0.898)] was a protective factor for dignity impairment symptoms in elderly

patients with Incontinence-Associated Dermatitis.See S2 Table. Assignment of Independent Variable Values, and S3 Table. Multifactorial Analysis Results of Dignity Impairment Symptoms in Elderly Patients with Incontinence-Associated Dermatitis.

## 4  Discussion

### 4.1  The incidence of dignity impairment symptoms in elderly patients with incontinence-associated dermatitis is at a medium-low level

This study surveyed 372 elderly patients with incontinence-associated dermatitis (IAD) across 42 tertiary hospitals in 11 provinces and autonomous regions in China. The results showed that approximately 35% of the patients experienced dignity impairment symptoms, reflecting a medium-low incidence rate.

In contrast to the 39% incidence of dignity impairment reported in ICU patients [15]. The difference can be attributed to the distinct clinical characteristics of the two patient groups. IAD typically occurs in the context of non-critical illnesses, which are more controllable and have better treatment outcomes. Consequently, patients with IAD tend to have a stronger sense of control over their condition and experience less psychological stress. In contrast, ICU patients often face critical conditions, accompanied by multiple comorbidities and functional impairments. Many of these patients, due to issues such as enteral nutrition intolerance or diarrhea, are prone to developing IAD [16]. In these cases, IAD not only exacerbates physical discomfort but also induces pain and frequent caregiving, which can further harm the psychological state and sense of dignity of the patients.

Although the incidence of dignity impairment symptoms in patients with IAD is relatively low, the psychological and quality-of-life impacts should not be overlooked. This is particularly important for elderly patients, for whom IAD not only increases physical discomfort but may also intensify psychological burdens, adversely affecting overall health. Due to both physiological and psychological factors, elderly patients tend to be more sensitive to incontinence issues and more likely to feel a loss of dignity. Therefore, clinical nursing care for IAD should pay closer attention to the psychological needs of these patients. Comprehensive measures should be adopted to alleviate physical symptoms and provide psychological support, through both counseling and compassionate care, which can effectively reduce the incidence of dignity impairment and improve the quality of life.

### 4.2  Factors influencing dignity impairment symptoms in elderly patients with incontinence-associated dermatitis

The results of this study indicate that female patients have a lower risk ($OR = 0.488$) of dignity impairment compared to male patients. Possible reasons for this finding include: Female patients with incontinence-associated dermatitis often receive more social support and emotional care, which helps them better cope with the psychological and physical challenges associated with the condition [17]. Additionally, societal and familial expectations of women typically involve more caregiving and support roles, which may reduce feelings of shame and frustration when facing illness, thereby lowering the risk of dignity impairment.

In contrast, male patients may experience a higher risk of dignity impairment due to societal expectations of strength and independence [18]. These expectations may lead male patients to conceal symptoms or delay seeking help when dealing with conditions like incontinence-associated dermatitis, thus increasing their risk of dignity impairment [19]. Therefore, it is recommended that clinical healthcare providers offer enhanced emotional support and psychological interventions for male patients, particularly when addressing similar conditions. This approach can help reduce psychological stress and improve disease

management. Targeted psychological support strategies, such as specific health education and counseling tailored for male patients, should be considered to help them address their illness more proactively and improve their overall quality of life.

This study found that employment status ($OR = 4.183$) is statistically significant in relation to dignity impairment. Specifically, retired patients have a higher risk of dignity impairment compared to employed patients. The following factors may explain this finding:Retired patients may experience increased psychological and social adjustment pressures when facing health issues such as incontinence-associated dermatitis. The transition to retirement often involves a reduction in social roles and activities, which may lead to decreased social support and increased feelings of isolation and psychological burden [20]. Additionally, economic pressures and declines in quality of life associated with retirement may negatively impact their sense of dignity [21]. These factors combined may contribute to a higher risk of dignity impairment among retired patients.

To mitigate the risk of dignity impairment in retired patients, it is recommended that clinical practices include enhanced psychological support and social interventions. For example, providing more opportunities for social engagement, such as group activities or community programs, can help retired patients build supportive social networks and improve their ability to cope with health challenges. Counseling services should also be available to address emotional distress and help manage feelings of isolation or depression.

Additionally, tailored health management plans should be considered, including economic support and quality of life improvement measures. Financial counseling could assist patients in managing post-retirement economic challenges, while access to affordable healthcare resources or subsidies for medical supplies can ease the burden of health issues [22]. These interventions can help retired patients better adapt to health challenges and preserve their dignity.

This study found that the type of primary caregiver ($OR = 1.451$) is statistically significant in relation to dignity impairment in patients with incontinence-associated dermatitis. Specifically, the risk of dignity impairment varies with the closeness of the caregiver relationship, ranging from spouses to children to others.The analysis revealed that patients whose primary caregivers are spouses experience the lowest risk of dignity impairment. This is likely because spouses, being the closest family members, can provide more comprehensive and compassionate care [23]. In contrast, patients with children as primary caregivers have a higher risk of dignity impairment, as children, despite their close relationship, may face challenges such as caregiver burden and limitations in managing the condition effectively [24]. Patients with more distant caregivers, such as other relatives or friends, have the highest risk of dignity impairment due to more distant emotional support and potential limitations in caregiving quality.

These findings suggest that the closeness of the caregiver relationship significantly impacts the risk of dignity impairment. To mitigate this risk, it is recommended to actively encourage the involvement of close family members, such as spouses and children, in the caregiving process. Since these relatives are more closely connected to the patient, they can provide more effective emotional support and care. Enhancing training and support for these caregivers and promoting their active participation in care can improve the patient's overall care experience and sense of dignity, thereby reducing the risk of dignity impairment.

This study found that self-reported family harmony ($OR = 0.703$) is a protective factor against dignity impairment in patients with incontinence-associated dermatitis. Specifically, patients who rate their family relationships as more harmonious have a lower risk of dignity impairment.The analysis indicates that patients with positive evaluations of their family relationships generally experience lower risks of dignity impairment. This is likely due to the

emotional support and psychological relief provided by harmonious family relationships. When family members maintain good relationships, patients are better able to cope with the psychological stress and emotional challenges associated with their illness and treatment, which in turn reduces the incidence of dignity impairment [25]. A supportive and positive family environment contributes to the patient's overall well-being and enhances their acceptance of care and treatment.

To further reduce the risk of dignity impairment, it is recommended to promote family harmony. Encouraging open communication among family members is crucial, and regular family meetings can help address conflicts and improve mutual understanding and support. Additionally, providing psychological counseling and support services to both patients and their families can help manage relationship challenges and enhance interaction quality. Training family members in disease management and caregiving skills is also important, as it improves their confidence and ability to support the patient effectively. Finally, actively involving family members in the patient's care plan and ensuring they are informed about the patient's condition and care needs can strengthen overall caregiving capacity and enhance the patient's sense of dignity [26]. By implementing these measures, family harmony can be improved, thereby reducing the risk of dignity impairment.

For the elderly population, patients with primary respiratory ($OR = 5.053$) or nervous system ($OR = 2.452$) diseases exhibit a significantly higher risk of dignity impairment compared to those with digestive system diseases. Chronic respiratory and neurological disorders are common among elderly individuals, characterized by long-term recurrence and persistence, which often leads to prolonged physical discomfort and functional limitations [27]. Patients with chronic respiratory and neurological diseases frequently face persistent symptoms and functional impairments, adversely affecting not only their physical health but also their psychological well-being. The enduring nature of these conditions can contribute to a sense of helplessness and dependency, thereby increasing the risk of dignity impairment [28]. Persistent pain and functional limitations lead to greater dependence on others in daily life, further undermining the patient's sense of self-worth and dignity.

To mitigate the risk of dignity impairment in these patients, several measures are recommended. First, comprehensive disease management and personalized care should be provided, including symptom control, functional rehabilitation, and psychological support. For patients with chronic respiratory and neurological diseases, essential interventions such as rehabilitation training, assistive devices, and support for daily living should be offered to alleviate physical discomfort and functional impairments. Additionally, enhancing psychological support and emotional care is crucial to help patients cope with the psychological stress associated with long-term illness, thereby improving their quality of life and sense of dignity. Encouraging patient and family involvement in disease management and rehabilitation can also enhance the patient's sense of self-efficacy and control over their life [29]. Implementing these measures can effectively improve the care experience for elderly patients and reduce the incidence of dignity impairment.

## 5 Conclusion

This study evaluated 372 elderly patients with incontinence-associated dermatitis across 42 tertiary hospitals in China, revealing that approximately 35% of the patients experienced varying degrees of dignity impairment. Key factors associated with dignity impairment included patient age, primary disease type, caregiver relationship, and self-reported family harmony. The findings suggest that dignity impairment is significantly influenced by both the burden of chronic disease and the nature of familial and caregiving relationships. Future research should focus on exploring effective interventions and support strategies for enhancing dignity

in elderly patients, particularly emphasizing the role of family involvement and the impact of chronic disease management. A multidisciplinary approach is recommended to develop comprehensive care models that address the psychological, social, and medical aspects of dignity preservation in this patient population.

## 6 Limitations

Although we made efforts to include a diverse sample, selection bias may still exist in this study. The distribution of questionnaires across multiple institutions may have unintentionally excluded or underestimated certain patient populations. This limitation suggests that the findings of our study may not be fully generalizable to all populations experiencing dignity impairment. To mitigate this, conducting a multicenter randomized sampling survey in future research could enhance the representativeness of the sample, address potential selection biases, and improve the generalizability of the findings. Additionally, this study employed a cross-sectional design, which provides data at a single time point and prevents the determination of temporal relationships between variables, making it difficult to establish causal links. Future research could adopt a longitudinal design to track patients over time and more accurately assess the causal relationships between risk factors and dignity impairment.

## Supporting information

**S1 Fig. Lasso coefficient path for variables in assessing dignity impairment symptoms in elderly patients with incontinence-associated dermatitis.**
(DOCX)

**S1 Table. Variables selected by LASSO analysis for assessing dignity impairment symptoms in elderly patients with incontinence-associated dermatitis.**
(DOCX)

**S2 Table. Assignment of independent variable values.**
(DOCX)

**S3 Table. Multifactorial analysis results of dignity impairment symptoms in elderly patients with incontinence-associated dermatitis.**
(DOCX)

## Author contributions

**Conceptualization:** jinlei du, xiaoling wu.

**Data curation:** jinlei du, xiaoling wu, ling lei, hongxiang zhao, qiyu zhang, yuanxia wang, yao chen, chencong nie, jiquan zhang.

**Formal analysis:** ling lei, jiquan zhang.

**Funding acquisition:** xiaoling wu, ling lei.

**Investigation:** jinlei du, xiaoling wu, ling lei, hongxiang zhao, yuanxia wang, yao chen.

**Methodology:** jinlei du, hongxiang zhao, yuanxia wang, yao chen, chencong nie.

**Project administration:** xiaoling wu, chencong nie.

**Resources:** jinlei du, xiaoling wu, qiyu zhang, chencong nie, jiquan zhang.

**Software:** jinlei du, ling lei, qiyu zhang, chencong nie.

**Writing – original draft:** jinlei du.

**Writing – review & editing:** jinlei du.

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
