## [Decision Letter · Decision Letter 0]

22 Dec 2024

PONE-D-24-35370Univariate Screening and Multivariate Analysis of Factors Influencing Dignity Impairment in Elderly Patients with Incontinence-Associated Dermatitis: A Lasso and Logistic Regression ApproachPLOS ONE

Dear Dr. nie,

Thank you for submitting your manuscript to PLOS ONE. After careful consideration, we feel that it has merit but does not fully meet PLOS ONE’s publication criteria as it currently stands. Therefore, we invite you to submit a revised version of the manuscript that addresses the points raised during the review process.

**ACADEMIC EDITOR:**

Address the following comments

This are my comments to the authors

General comments

It needs grammar edition (capitalization,…)

Title should be modified as - ***Factors Influencing Dignity Impairment in Elderly Patients with Incontinence-Associated Dermatitis: A Lasso and Logistic Regression Approach*****Limited keywords- add more  words**

**
*Introduction*
**

In line 35 a statement: This condition can affect skin areas beyond the perineum. Have no referenceStatement in line 45-47 - Current research both domestically and internationally mainly focuses on the 46 prevention and treatment of IAD symptoms, with limited studies addressing 47 psychological aspects, especially in the domain of dignity. **Cite reference**

**
*General: the introduction part should be clearly stated and written with sufficient information*
**

***Problem statement of Dignity Impairment******Potential significance of your study******Use appropriate reference***

**
*Materials and Method*
**

In line 85: Exclusion criteria: you stated that history of depression within the past three months; diagnosis 86 of psychiatric disorders; presence of other severe, incurable chronic diseases; 87 participation in concurrent psychological intervention studies. Why 3 months? Do you mean that psychiatric disease onset within three months does not affect the results?

Line 88: This study has been 88 approved by the hospital’s ethics committee: remove from this part

**Discussion**

Avoid subtitle in the discussion part

The discussion part is lacking, it needs more scientific justification

We look forward to receiving your revised manuscript.

Kind regards,

Bedilu Linger Endalifer

Academic Editor

PLOS ONE

3. Please amend your authorship list in your manuscript file to include author jinlei du, xiaoling wu, ling lei, hongxiang Zhao, qiyu zhang, yuanxia wang, yao chen, jiquan zhang, chencong nie.

Additional Editor Comments (if provided):

Reviewers' comments:

Reviewer's Responses to Questions

**Comments to the Author**

1. Is the manuscript technically sound, and do the data support the conclusions?

Reviewer #1: Partly

Reviewer #2: Partly

Reviewer #3: Yes

Reviewer #4: Partly

2. Has the statistical analysis been performed appropriately and rigorously?

Reviewer #1: Yes

Reviewer #2: Yes

Reviewer #3: Yes

Reviewer #4: No

3. Have the authors made all data underlying the findings in their manuscript fully available?

Reviewer #1: Yes

Reviewer #2: No

Reviewer #3: Yes

Reviewer #4: Yes

4. Is the manuscript presented in an intelligible fashion and written in standard English?

Reviewer #1: Yes

Reviewer #2: Yes

Reviewer #3: Yes

Reviewer #4: Yes

5. Review Comments to the Author

Reviewer #1: Here are my comments on the key aspects.

Technical soundness and data support:

The manuscript is partly technically sound, with data that partly supports the conclusions. The study addresses an important topic with an adequate sample size and appropriate statistical methods. However, there are some limitations.

-The cross-sectional design limits causal inferences about risk factors.

-There's limited information on the validity of the dignity assessment scale used.

-Some potential confounding factors (e.g. IAD severity, comorbidities) appear not to have been controlled for.

-Addressing these issues would strengthen the conclusions.

Statistical analysis:

The statistical analysis appears appropriate and rigorous. It is appropriate to use Lasso regression followed by logistic regression to identify risk factors. The sample size calculation and reporting of odds ratios with confidence intervals are appropriate. However, more details on the specific statistical procedures would be beneficial.

Manuscript presentation:

The manuscript is clearly written in standard English, with a logical structure that effectively communicates the study's objectives, methods, results, and conclusions.

Additional comments:

-Provide more details on how IAD was diagnosed and graded.

-Clarify the validity and reliability of the dignity assessment scale used.

-Discuss potential confounding factors not accounted for in the analysis.

Overall, this study provides valuable initial data on an important topic, but addressing the noted limitations would significantly strengthen the manuscript.

Reviewer #2: There is no data on the functional status of the subjects .How many had poor functional status ? Dignity is proportional to independent status .Also gender of care giver ,whether care giver was a hired care giver ,spouse or daughter or son or daughter in law or a qualified nurse is important to know .Such a high prevalence of dermatitis has been reported .Were subjects using incontinence solutions such as diapers or leak proof pants ?

How was the cognitive status of elders who answered the dignity questionnaire ?were they literate or illiterate

Very often dermatitis is due to candidal intertrigo which is curable with antifungal agents .Was there pain which impaired quality of life

comparison of dignity questinnaire of incontinent people with dignity of a patient suffering from cancer is not appropriate

Reviewer #3: The research article exhibits several methodological and presentation-related errors that warrant attention. Firstly, the use of convenience sampling introduces selection bias, potentially limiting the generalizability of the findings. While the LASSO regression approach is appropriate, details on hyperparameter tuning and comprehensive multicollinearity diagnostics (e.g., variance inflation factor values) are missing, reducing the study's reproducibility. The sample size slightly exceeds the calculated requirement without a clear explanation, and potential confounders, such as education level and urban/rural residence, are not fully addressed. Furthermore, the discussion lacks depth in contextualizing findings, particularly in offering actionable interventions for identified protective and risk factors. Ethical considerations, while mentioned, do not detail measures taken to mitigate psychological distress during data collection. Additionally, vague statements about data availability ("on reasonable request") hinder transparency, and referenced figures and tables are absent in the manuscript, impacting the clarity of the presented results. Minor grammatical inconsistencies and overly complex sentences further detract from the article's readability and coherence.

Reviewer #4: It is not prudent to specify a statistical test in the title, even if it may be informative.

In the introduction, the importance of the study is not adequately specified. It is important to expand the topic and references based on points such as the management of dermatological diseases in older adults, the quality of life in people with urinary incontinence.

The study design is not defined.

It is understood that if the study is analytical according to the title, a sample size for only one population should not be used.

In the exclusion criteria, it is not specified what types of incurable or severe diseases, the presence or absence of disability or functional impairment, and whether they had any other type of dermatological condition.

It is not specified when talking about the dignity test the cutoff point for evaluating its deterioration.

In the statistical section, they detail the type of analysis and the procedure for a multivariate analysis. It is also important to detail the assumptions for conducting this analysis and whether all of them were met.

Urinary incontinence is not evaluated as a confounding variable because it can itself alter the patient's dignity, as the test assesses dignity in a global manner and not defined by diseases.

In the discussion: the comparison should ideally be with chronic and/or dermatological pathologies in this population.

Explain how they addressed the limitations and strengths of the study.

6. PLOS authors have the option to publish the peer review history of their article (what does this mean? ). If published, this will include your full peer review and any attached files.

**Do you want your identity to be public for this peer review?** For information about this choice, including consent withdrawal, please see our Privacy Policy .

Reviewer #1: **Yes: ** RICHARD AMOAKO

Reviewer #2: No

Reviewer #3: **Yes: ** Salman Ashfaq Ahmad

Reviewer #4: No

---

## [Author Response · Author response to Decision Letter 1]

15 Jan 2025

Dear Editor,Thank you for the valuable guidance and revision suggestions provided by you and the reviewers. We have uploaded all our responses in the attached file, named "Response to Editor and Reviewers."

---

## [Editor Report · Decision Letter 1]

18 Feb 2025

Factors Influencing Dignity Impairment in Elderly Patients with Incontinence-Associated Dermatitis: A Lasso and Logistic Regression Approach

PONE-D-24-35370R1

Dear Dr. du,

We’re pleased to inform you that your manuscript has been judged scientifically suitable for publication and will be formally accepted for publication once it meets all outstanding technical requirements.

Kind regards,

Bedilu Linger Endalifer

Academic Editor

PLOS ONE
---

## [Editor Report · Acceptance letter]

PONE-D-24-35370R1

PLOS ONE

Dear Dr. du,

I'm pleased to inform you that your manuscript has been deemed suitable for publication in PLOS ONE. Congratulations! Your manuscript is now being handed over to our production team.

Kind regards,

on behalf of

Dr. Bedilu Linger Endalifer

Academic Editor

PLOS ONE